# Grokking in the Wild: Data Augmentation for Real-World Multi-Hop Reasoning with Transformers

**Roman Abramov** [1]  **Felix Steinbauer** [1]  **Gjergji Kasneci** [1 2]

## Abstract

Transformers have achieved great success in numerous NLP tasks but continue to exhibit notable gaps in multi-step factual reasoning, especially when real-world knowledge is sparse. Recent advances in grokking have demonstrated that neural networks can transition from memorizing to perfectly generalizing once they detect underlying logical patterns – yet these studies have primarily used small, synthetic tasks. In this paper, for the first time, we extend grokking to real-world factual data and address the challenge of dataset sparsity by augmenting existing knowledge graphs with carefully designed synthetic data to raise the ratio $\phi_r$ of inferred facts to atomic facts above the threshold required for grokking. Surprisingly, we find that even factually incorrect synthetic data can strengthen emergent reasoning circuits rather than degrade accuracy, as it forces the model to rely on relational structure rather than memorization. When evaluated on multi-hop reasoning benchmarks, our approach achieves up to 95–100% accuracy on 2WikiMultiHopQA – substantially improving over strong baselines and matching or exceeding current state-of-the-art results. We further provide an in-depth analysis of how increasing $\phi_r$ drives the formation of generalizing circuits inside Transformers. Our findings suggest that grokking-based data augmentation can unlock implicit multi-hop reasoning capabilities, opening the door to more robust and interpretable factual reasoning in large-scale language models.

---

[1]School of Computation, Information and Technology, Technical University of Munich, Munich, Germany [2]School of Social Sciences and Technology, Technical University of Munich, Munich, Germany. Correspondence to: Roman Abramov <roman.abramov@tum.de>, Felix Steinbauer <felix.steinbauer@tum.de>.

*Proceedings of the $42^{nd}$ International Conference on Machine Learning*, Vancouver, Canada. PMLR 267, 2025. Copyright 2025 by the author(s).

## 1  Introduction and Related Work

Transformers have demonstrated remarkable success across a wide range of natural language processing (NLP) tasks, such as text classification, summarization, and machine translation. Nevertheless, they still face significant challenges when asked to perform *multi-step* or *multi-hop* factual reasoning, particularly in real-world scenarios where knowledge is both vast and sparsely distributed. A key reason for this difficulty lies in the model's tendency to memorize rather than *generalize* – a problem that becomes acute in knowledge-intensive tasks with insufficiently rich data distributions.

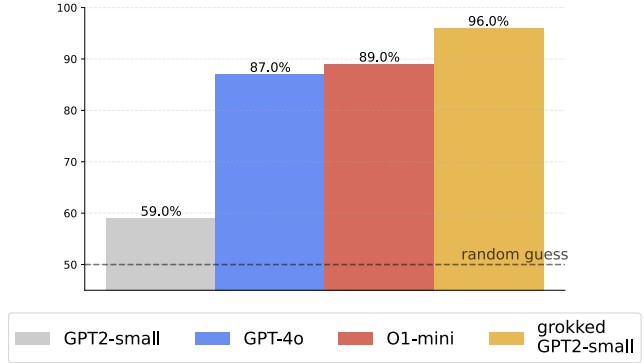

*Figure 1.* Average accuracy on 2WikiMultiHopQA for comparison task. Despite GPT2-small being a model with 124 million parameters, grokked version achieves almost 100% accuracy, beating the most recent gpt-4o and o1-mini models.

**From Toy Grokking to Real-World Data.** Recent work on *grokking* (Power et al., 2022) has shown that, under certain conditions, overparameterized neural networks suddenly transition from pure memorization to near-perfect generalization after long training. Early studies have typically focused on highly controlled, synthetic tasks such as modular arithmetic or simplified algorithmic datasets. In these *toy* settings, the number of *inferred facts* (i.e., multistep or composed patterns) can be systematically increased until a threshold ratio $\phi$ is reached, at which point a "generalizing circuit" emerges in the model (Belkin et al., 2019; Nakkiran et al., 2020; Thilak et al., 2022; Humayun et al., 2023; Nanda et al., 2023).

However, real-world datasets present a stark contrast: fac-

tual knowledge is *extremely sparse* and often scattered across incomplete or noisy knowledge graphs. Thus, the core challenge is to ensure that there are enough higher-order *inferred facts* in relation to the *atomic facts* (direct statements) to enable the internal circuit-formation process that grokking requires. Put differently, in real-world scenarios, one cannot trivially guarantee a sufficiently large ratio $\phi$ between multi-step (inferred) facts and single-hop (atomic) facts. Our work addresses precisely this obstacle by proposing a data-synthesis strategy that augments and re-balances real-world knowledge bases.

**Multi-hop Question Answering (QA).** 2WikiMulti-HopQA (Yang et al., 2018; Ho et al., 2020; Trivedi et al., 2022) is particularly well-suited to assess multi-step factual reasoning. This dataset contains Wikipedia-based queries that require retrieving and combining multiple pieces of evidence (spread across different pages or paragraphs) before producing an answer. This property aligns closely with our focus on multi-hop reasoning and underscores the need for implicit reasoning capability: in many cases, a system must link and traverse *several* factual nodes – e.g., "Michelle is the wife of Obama" and "Michelle was born in 1964" – to arrive at a final conclusion (e.g., about Michelle's birth year or other derived facts). Moreover, they reflect a large, complex knowledge graph with real-world entities, ambiguous language, and long-tail relations – all of which make them an archetypal testbed for factual reasoning at scale.

**Grokking and Its Role in Transformer Generalization.** The concept of grokking, introduced in Power et al. (2022), demonstrated that neural networks could learn not just superficial patterns but also deeper, generalizable reasoning mechanisms under prolonged training with suitable inductive biases. Subsequent work has linked grokking to double descent (Belkin et al., 2019; Nakkiran et al., 2020), the geometry of deep-network loss landscapes (Davies et al., 2023), and weight decay (Pezeshki et al., 2022; Nanda et al., 2023), suggesting that the right regularization can encourage the emergence of *generalizing circuits*. These circuits – once formed – enable out-of-distribution reasoning that surpasses naive memorization (Varma et al., 2023; Liu et al., 2023).

**Gap in the Literature.** Despite extensive research on *knowledge graph completion* (Liu et al., 2022) and multi-hop question answering via retrieval-based methods (Yang et al., 2018; Ho et al., 2020), very few studies have examined whether the *internal grokking phenomenon* can be harnessed for implicit multi-hop reasoning in a *real-world* textual setting. Most prior approaches either:

- **Focus on toy tasks**: e.g., modular addition or synthetic math problems (Power et al., 2022; Nanda et al., 2023; Wang et al., 2024).
- **Rely on explicit prompting or chain-of-thought**: where

intermediate reasoning steps must be spelled out in external text (Plaat et al., 2024), rather than learned as an implicit circuit.
- **Use standard graph-completion architectures**: e.g., GNN-based solutions to augment partial knowledge graphs (Liu et al., 2022), which do not necessarily yield late-phase *internal* circuit formation or sudden generalization.

To the best of our knowledge, no existing work has used a *grokking-based* approach to demonstrate how a Transformer can *implicitly* discover multi-hop reasoning skills on large-scale factual data.

**Contributions.** In this paper, we bridge that gap through the following contributions:

- We incorporate a targeted *data synthesis* procedure to ensure sufficiently large $\phi$ for each relation, thereby unlocking the potential for *internal generalization circuits* to form in real-world Wikipedia-based tasks.
- We show that *even factually incorrect synthetic data can boost the ratio of inferred to atomic facts*, often strengthening rather than harming logical consistency.
- Our experiments on 2WikiMultiHopQA confirm that once the ratio $\phi$ surpasses a certain threshold, grokking emerges – enabling Transformers to perform complex multi-step reasoning *without* explicit chain-of-thought prompts or elaborate external scaffolding.

In the following sections, we detail the mathematical basis of our data augmentation strategy, the empirical setups on 2WikiMultiHopQA, and the new insights gained about implicit multi-hop reasoning. Our findings show that *grokking is not an artifact confined to contrived toy datasets* but a powerful mechanism that, with suitable data distribution adjustments, can be harnessed for real-world factual reasoning at scale.

## 2 Problem Description

The concept of multi-hop reasoning presupposes a knowledge graph (KG) whose nodes (entities) and edges (relations) can be traversed via a chain of inference steps. In our setting, this KG is encoded in *textual* form, but structurally, we are still dealing with *multi-hop question answering* over a KG (Multi-hop KGQA). Prior work (Liu et al., 2022) has already observed that *knowledge graph completion* can be critical for multi-hop KGQA; for *grokking*-based Transformer generalization circuits to form, it becomes *imperative* to extend the original KG such that sufficiently many multi-step (inferred) facts exist. This section formalizes our problem of *augmenting* a KG to enable Transformer grokking.

## 2.1 Definitions and Basics

We begin by introducing key notations for clarity. Readers can refer to Table 1 at any time for a concise summary of the main symbols.

**Knowledge Graph.**

**Definition 2.1** (Knowledge Graph). We define a knowledge graph as a tuple $\mathcal{KG} = (\mathcal{V}, \mathcal{R}, \mathcal{F}_A)$, where

- $\mathcal{V}$ is a finite set of **entities** (nodes or vertices),
- $\mathcal{R}$ is a finite set of **relation types** (edges/predicates),
- $\mathcal{E} \equiv \mathcal{F}_A \subseteq \mathcal{V} \times \mathcal{R} \times \mathcal{V}$ is a finite set of **atomic facts** (triplets) of the form $(h, r, t)$, where $h \in \mathcal{V}$ is a *head* (subject), $t \in \mathcal{V}$ is a *tail* (object), and $r \in \mathcal{R}$ is the relation.

In natural language, entities are typically connected by relational statements. Although some relations (e.g., "son of") are directed, one can also treat the KG as undirected for *graph traversal* since a statement is often queryable in both directions (subject/object).

**Definition 2.2** (Norm over edges). For counting strictly *directed* edges in $\mathcal{KG}$ *as if* they were undirected, we use the trivial count: $| \; | = \sum_{(h,\dots,t) \in \mathcal{E}} 1$.

**Definition 2.3** (Average branching factor). The *average branching factor* of a knowledge graph is: $b = \frac{|\mathcal{F}_A|}{|\mathcal{V}|}$.

**Definition 2.4** (Atomic Facts). Following prior work, we use $\mathcal{F}_A$ (or equivalently $\mathcal{F}_1$) to denote the set of **atomic facts**, i.e., all **first-order** triplets explicitly stored in the knowledge graph.

**Example 1** (Running Example: Basic KG). *Consider the KG in Figure 2 with*

$\mathcal{V} = \{$"*Obama*", "*Michelle*", "*1964*", "*Mary Poppins*"$\}$,
$\mathcal{R} = \{$"*wife of*", "*born in*", "*aired in*"$\}$.

*The atomic facts* $\mathcal{F}_A$ *include:*

$$\big(\text{"Michelle", "wife of", "Obama"}\big),$$
$$\big(\text{"Michelle", "born in", "1964"}\big),$$
$$\big(\text{"Mary Poppins", "aired in", "1964"}\big).$$

*Hence,* $\mathcal{F}_1 = \mathcal{F}_A$. *The average branching factor here is* $b = 0.75$.

**Inference Steps and Paths.**

**Definition 2.5** (Inference Step). An inference step is a function $I : \mathcal{V} \times \mathcal{R} \to \mathcal{V}$ that traverses the graph from one entity to a neighboring entity via one relation:

$$I(h, r) = t \quad \text{where} \quad (h, r, t) \in \mathcal{F}_A.$$

**Definition 2.6** (Inference Path). An inference path $p_n$ of length $n$ is a sequence of relations $p_n = (r_1, \dots, r_n) \in \mathcal{R}^n$. The path is **simple** if each relation leads to exactly one

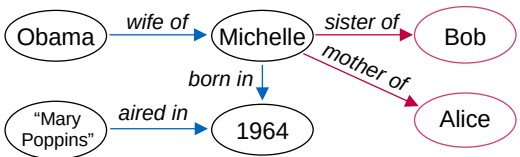

*Figure 2.* **Exemplary Knowledge Graph with Synthesized Data.** Four original nodes (**black**) and three relations (**blue**) result in two inferred facts. Two additional synthetic nodes and relations (**red**) extend the amount of inferred facts by four. Consequently, $\phi_I = \phi_2$ increases from $\frac{2}{3} \approx 0.66$ to $\frac{6}{5} = 1.2$.

successor node (no branching ambiguity):

$$\forall r_i \in p_n \; : \exists v_{i-1}, v_i \in \mathcal{V} \text{ such that } I(v_{i-1}, r_i) = v_i$$
$$\wedge \, \forall v_j \in \mathcal{V} : \big( I(v_{i-1}, r_i) = v_j \implies v_j = v_i \big).$$

**Definition 2.7** (N-th Order Deductions). For an inference path $p_n = (r_1, \dots, r_n)$ of length $n$ connecting $v_0$ (head) to $v_n$ (tail), the **n-th order deductions** are:

$$\mathcal{F}_n \subseteq \big\{ (v_0, r_1, \dots, r_n, v_n) \mid v_i \in \mathcal{V}, r_i \in \mathcal{R} \big\}.$$

**Example 2** (2-Hop Deductions). *A second-order (2-hop) fact arises from concatenating two atomic facts via one bridge entity. For instance:*

$$(h, r_1, b, r_2, t) \quad \text{with} \quad (h, r_1, b), (b, r_2, t) \in \mathcal{F}_A.$$

*In Example 1, we might ask:* "Which year was Obama's wife born in?" $\Rightarrow I($"*Obama*", "*wife of*"$) = $"*Michelle*" *and* $I($"*Michelle*", "*born in*"$) = $"*1964*".

**Inferred vs. Atomic Facts.**

**Definition 2.8** (Inferred Facts). All **n-th order** facts with $n > 1$ constitute the *inferred facts*,

$$\mathcal{F}_I = \bigcup_{n=2}^{\infty} \mathcal{F}_n.$$

Clearly, $\mathcal{F}_A \cap \mathcal{F}_I = \emptyset$.

**Example 3** (3-Hop Deduction). *Continuing the example of Obama, consider the path* $p_3 = ($"*wife of*", "*born in*", "*aired in*"$)$. *By chaining three steps, we deduce:*

$$(\text{"Obama", "wife of", "born in", "aired in", "Mary Poppins"})$$

*which answers* "Which movie aired in the same year Obama's wife was born?"

## 2.2 Generalization Over Inference Paths

**Definition 2.9** (Implicit Reasoning). We define *implicit reasoning* as reasoning *without* the need for explicit intermediate prompts or developer-introduced structure.

Unlike *explicit* reasoning approaches (e.g., chain-of-thought prompts (Plaat et al., 2024)), *implicit* reasoning relies on the model forming *internal* circuits during training (Nanda

et al., 2023). Grokking studies (Power et al., 2022; Wang et al., 2024) have shown that Transformers can, under certain conditions, shift from memorizing to perfectly generalizing by learning these internal circuits.

**Relation-Specific Ratios.** As noted by Wang et al. (2024), a critical factor for circuit formation is the ratio of *inferred facts* to *atomic facts* for each relation $r$. Let

$$\mathcal{F}_{A,r} \subseteq \mathcal{F}_A \quad \text{and} \quad \mathcal{F}_{I,r} \subseteq \mathcal{F}_I$$

denote the sets of atomic and inferred facts that *involve* relation $r$. Then:

**Definition 2.10** (Generalization Ratio). For each relation $r \in \mathcal{R}$, we define $\phi_r = \dfrac{\left|\mathcal{F}_{I,r}\right|}{\left|\mathcal{F}_{A,r}\right|}$.

When $\phi_r$ crosses a certain threshold—empirically found to be around 3.6 for slow generalization and up to 18 for faster circuits in GPT-2 style Transformers—the model tends to form a *generalizing circuit* for that relation(Power et al., 2022). These thresholds are approximate and architecture-dependent (Nanda et al., 2023; Wang et al., 2024) but illustrate the core principle: *without sufficiently many multi-hop facts, the model never "grokks" the underlying relation.*

**Partial vs. Full Generalizability.** Even if a knowledge base is only *partially* rich in multi-hop data (i.e., some relations meet the threshold $\phi_G$, while others do not), it can still benefit certain reasoning tasks. However, to achieve *full* generalizability, *all* relations must surpass the threshold:

**Definition 2.11** (Generalizable Knowledge Base). A knowledge base $\mathcal{KG}$ is **generalizable** if

$$\forall r \in \mathcal{R} \colon \phi_r \ \geq \ \phi_G,$$

where $\phi_G$ is the *minimal generalization ratio* required by a given model setup. If this condition holds for only a subset of relations, we call $\mathcal{KG}$ *partially generalizable*.

### 2.3 Bounds for Generalizable Knowledge Bases

Because each relation $r$ has its own ratio $\phi_r$, a knowledge base (KB) may fail to support grokking if any $\phi_r$ remains too low. Analytic bounds (see Appendix A.3) reveal that:

- The ratio $\phi_r$ can be increased by adding new nodes or by augmenting edges related to $r$, but this effect is limited by how the graph is connected and how often the relation $r$ appears.
- Typical real-world KGs tend to be *sparse*, which is why data synthesis (described in later sections) is crucial to boost $\phi_r$.

**Example 4** (Insufficient Branching Factor). *Suppose a family KG has $\{father, mother, child_1, \ldots\}$ with edges like "parent of," "sibling of," "owned by." Although the graph is well-connected at a glance, its relation-specific branching factors might still be too small to pass the threshold*

$\phi_G \approx 3.6$. *Hence, the KB remains not fully generalizable, preventing a Transformer from forming robust multi-hop circuits for all relations.*

### 2.4 Setup and Querying

In practice, we test a model's multi-hop reasoning by asking queries whose *unique answer* resides at the end of a *simple* (acyclic) inference path. For instance:

**Example 5** (Chaining 3 Steps). *The textual query "Which movie aired in the same year as Obama's wife was born?" corresponds to a 3-hop deduction:*

$$(\text{"Obama"}, \text{"wife of"}, \text{"born in"}, \text{"aired in"}, t).$$

*Here, the model must internally infer:*

$$I(\text{"Obama"}, \text{"wife of"}) \ \rightarrow \ \text{"Michelle"},$$
$$I(\text{"Michelle"}, \text{"born in"}) \ \rightarrow \ \text{"1964"},$$
$$I(\text{"1964"}, \text{"aired in"}) \ \rightarrow \ t = \text{"Mary Poppins"}.$$

*Successful* implicit *reasoning requires the model to* both memorize *the atomic facts and* chain them together *via a generalizing circuit.*

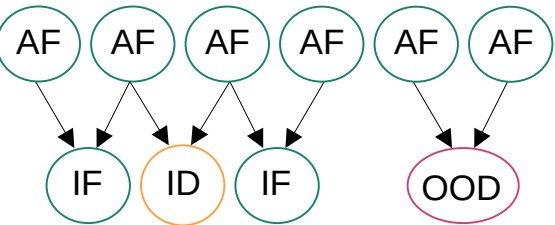

*Figure 3.* **Conceptual difference between ID and OOD:** In-distribution (**ID, 3orange**) and Out-of-distribution (**OOD, red**) inferred facts are shown. All **green** components are seen during training, including all atomic facts (**AF**) and some inferred facts (**IF**).

**Definition 2.12** (In-distribution vs. Out-of-distribution). **In-distribution (ID)**: An **inferred fact** derived from combinations of atomic facts that were **present in the training** data but never appeared together in this specific combination. The model **has seen** all knowledge components **separately** and different reasoning combinations involving them, but not in this particular test arrangement.
**Out-of-distribution (OOD)**: An **inferred fact** derived from atomic facts that were **present in the training** data but **never used** in **any** train reasoning paths. The model has seen the individual knowledge components, but not how they should be applied in the context being tested.
**Figure 3** shows trained facts (atomic and inferred) related to ID and OOD testing facts.

As we show in our experiments, *OOD* queries can be especially challenging, requiring truly *structural* generalization rather than partial memorization. This is precisely where sufficiently high $\phi_r$ ratios become critical for enabling the

Transformers' *grokking*-driven jump from local memorization to robust multi-hop reasoning.

## 2.5 Lemmas and Bounds

The first lemma we introduce makes clear that (1) picking nodes vs. arranging them in paths vs. the existence of edges leads to an asymptotic upper bound, and (2) even large KBs do not grow $\phi_{n,r}$ beyond roughly $b^{n-1}$ without additional data augmentation.

**Lemma 1 (Asymptotic Bound on the Number of $n$-hop Paths).** *Consider a knowledge base (KB) with $|\mathcal{V}|$ entities, average branching factor $b$, and an (approximate) random-graph assumption that each potential* directed *edge between two distinct nodes is present with probability $\frac{b}{|\mathcal{V}|-1}$. Then, for large $|\mathcal{V}|$, the expected number of valid n-hop paths satisfies:*

$$|\mathcal{F}_n| \approx \binom{|\mathcal{V}|}{n+1}(n+1)!\left(\frac{b}{|\mathcal{V}|-1}\right)^n.$$

*Moreover, in the limit $|\mathcal{V}| \to \infty$, the relation-specific ratio $\phi_{n,r}$ (i.e., the ratio of n-hop to 1-hop facts for relation r) remains bounded above by $b^{n-1}$.*

For a *Proof of Lemma 1* see Appendix A.1. For a lower generalization bound on the number of nodes $|\mathcal{V}|$, see Appendix A.3. The second lemma makes clear that each relation's branching factor $b_r$ must be sufficiently large to surpass the empirical threshold $\phi_G$. If one or more relations fall below that threshold, full generalization across *all* relations will not occur – even though partial generalization might emerge for the higher- $b_r$ relations.

**Lemma 2 (Necessary Condition for Full Generalizability).** *Let $\phi_G$ be the minimal ratio required to trigger grokking-based generalization for a given model architecture. A KB is **fully generalizable** over $n$-hop facts only if $\forall\, r \in \mathcal{R}$*

$$\phi_{n,r} \geq \phi_G \quad \Rightarrow \quad b_r > \sqrt[n-1]{\frac{\phi_G\,|\mathcal{V}|\,(|\mathcal{V}|-1)^n}{\binom{|\mathcal{V}|}{n+1}(n+1)!}},$$

*where $b_r = \frac{|\mathcal{F}_{A,r}|}{|\mathcal{V}|}$ is the relation-specific branching factor. Equivalently, if any $b_r$ falls below this threshold, the KB cannot be fully generalizable.*

For a *Sketch of Proof of Lemma 2* see Appendix A.2.

## 3 Method

Our method follows a two-stage pipeline designed to enable *grokking* in real-world multi-hop reasoning tasks. First, we **augment** the original Wiki2Hop dataset with synthetic knowledge, increasing the ratio of multi-hop (*inferred*) to single-hop (*atomic*) facts. Second, we **train** a Transformer model for hundreds of thousands of steps, leveraging the late-phase generalization phenomenon characteristic of

*Table 1.* **Key Notation Table.** Frequently used symbols throughout this paper.

| Symbol | Meaning |
|---|---|
| $\mathcal{V}$ | Set of entities (nodes) in the KG |
| $\mathcal{R}$ | Set of relation types (edges) |
| $\mathcal{F}_A$ | Atomic facts (1-hop) $\subseteq \mathcal{V} \times \mathcal{R} \times \mathcal{V}$ |
| $\mathcal{F}_I$ | Inferred facts (multi-hop) $\bigcup_{n \geq 2} \mathcal{F}_n$ |
| $b$ | Average branching factor $\frac{|\mathcal{F}_A|}{|\mathcal{V}|}$ |
| $\phi_r$ | $\frac{|\mathcal{F}_{I,r}|}{|\mathcal{F}_{A,r}|}$ (ratio of inferred to atomic facts) |
| $\phi_G$ | Minimal generalization ratio for successful grokking |
| $I(h, r)$ | Inference step: from entity $h$ via relation $r$ to a neighbor |

grokking (Power et al., 2022; Wang et al., 2024).

### 3.1 Dataset

**2WikiMultiHopQA Overview.** We use the **2WikiMultiHopQA** dataset (Ho et al., 2020), a well-known benchmark for retrieval-augmented generation (RAG) and multi-hop QA. 2WikiMultiHopQA consists of Wikipedia paragraphs, supporting facts (triplets), and reasoning queries that often require chaining multiple pieces of evidence. Despite its breadth, the *initial* ratio $\phi \approx 0.5$ – that is, each two atomic fact spawn only one inferred fact – making it insufficient for grokking to emerge naturally.

**Structured vs. Unstructured.** We divide 2WikiMultiHopQA into two subsets:

- **Structured**: supporting facts are simplified into short triplets (e.g., `Paris -- country -- France`).
- **Unstructured**: supporting facts are embedded in full Wikipedia paragraphs, offering richer context but also more noise and complexity.

Within these subsets, we focus on two of the four main multi-hop tasks:

1. **Comparison**, which compares attributes of different entities (e.g., same location or release year),
2. **Composition**, which chains multiple relations to derive answers (e.g., who directed the sequel of a certain film?).

Our objective is to *raise* $\phi$ for both comparison and composition queries so that Transformers can develop *internal* reasoning circuits.

### 3.2 Data Augmentation

To increase the coverage of multi-hop questions, we systematically add both *atomic* and *inferred* facts via LLM-based generation. In each case, we ensure consistency with 2WikiMultiHopQA style, maintain balanced class distributions, and closely track $\phi$ so it exceeds known thresholds for grokking (Wang et al., 2024).

#### 3.2.1 COMPARISON TASK

In the comparison task, atomic facts (e.g., `City -- country -- X`) are paired to form questions about

---

**Algorithm 1** Augmentation algorithm for comparison

---
**Require:** loc_examples  // sample atomic facts
**Require:** detailed_examples   // extended paragraphs
 1: atomic ← *generate_locations*(loc_examples)
 2: **if** task_type == "full_text" **then**
 3:     atomic ← *detalize_locations*(atomic, detailed_examples)
 4: **end if**
 5: inferred ← *generate_inferred*(atomic)
 6: **return** atomic, inferred

---

shared attributes. Initially, we select 120 atomic facts and 60 inferred facts centered on geographic locations (India, France, the U.S., Canada, Russia). Using our augmentation strategy, we expand this to 1,000 atomic facts and 8,000 inferred facts, yielding $\phi_G = 8$ – well above the bare-minimum ratio of 3.6 reported by Wang et al. (2024) for slow grokking.

**Generating New Locations.** We first produce atomic facts describing additional cities or regions not present in the original set. For the structured version, these remain in the (City, country, Country) format. For unstructured data, we prompt a Large Language Model (LLM) to generate concise, Wikipedia-like paragraphs (e.g., "*Paris Louvre Museum: The Louvre is a world-famous art museum...*").

**Generating Inferred Examples.** Next, we create comparison-based queries by selecting two distinct location facts and prompting, e.g., "*Are Avignon Rocher des Doms and Paris Louvre Museum both located in the same country?*." Algorithm 1 shows the pseudo-code for this procedure, where *generate_inferred* systematically merges atomic facts into comparison questions:

### 3.2.2 COMPOSITIONAL TASK

Compositional tasks require linking multiple relations in sequence (e.g., Person -- spouse of -- X -- born in -- Year). We begin with 200 atomic facts and 100 inferred facts, ensuring no direct mention of dates as an answer, and expand to 800 atomic facts plus 5,000 inferred facts. This boosts $\phi_G$ to 6.25 – enough to trigger grokking dynamics in practice.

**Graph Processing.** We transform the textual facts into a graph representation, extracting nodes (entities) and edges (relations) via an LLM-based parser. We then enrich this graph by adding new atomic edges (avoiding cycles) and sampling multi-hop paths that yield additional inferred facts.

**Inferred Question Pentads.** Each multi-hop path of length two or three corresponds to a pentad $(obj_1, rel_1, obj_2, rel_2, obj_3)$. An LLM then converts these pentads into natural-language questions (e.g., "*Why did Randal Plunkett, 19th baron of Dunsany's father die?*"), approximating real 2WikiMultiHopQA style. Algorithm 2 details this approach:

---

**Algorithm 2** Augmentation algorithm for composition

---
**Require:** text  // original atomic facts in textual form
 1: graph ← *parse_graph*(text)
 2: atomic ← graph.*augment_atomic*()
 3: inferred ← graph.*augment_inferred*()
 4: inferred ← *diversify*(inferred)
 5: **return** atomic, inferred

---

**Example of Augmented Facts.** Table 2 provides a sample of how an atomic fact, a detailed version, and an inferred question look after augmentation. Note how the final question elegantly ties both locations to a yes/no query on whether they share the same country.

*Table 2.* Examples of augmented facts for the 2WikiMultiHopQA comparison task

| Type | Example |
|---|---|
| **Atomic Fact** | `Louvre Museum -- country -- France` |
| **Detailed Fact** | `Paris Louvre Museum:  The Louvre Museum is a world-famous art museum...` |
| **Inferred Question** | *Are Avignon Rocher des Doms and Paris Louvre Museum both located in the same country? (Answer: Yes)* |

After these augmentation steps, the resulting dataset attains a sufficiently high $\phi$. In the next section, we detail how we **train** a GPT-2 style Transformer on this enriched corpus, and how prolonged optimization reveals the hallmark late-phase jump in multi-hop reasoning accuracy.

## 4 Experiments

We evaluate our grokking-based approach on the **2WikiMultiHopQA** dataset (Ho et al., 2020), augmented to ensure a sufficiently large ratio $\phi$. We report results on both **structured** (4.3) and **unstructured** (4.4) subsets, as well as the **original** (unaugmented) data (4.2). Finally, we compare performance across all settings (4.5) and offer qualitative insights (4.6).

### 4.1 Setup

**Model and Training.** We train an 8-layer GPT-2–style Transformer (768 hidden units, 12 attention heads) *from scratch* using AdamW (**?**) with a learning rate of $5 \times 10^{-5}$, batch size of 512, and weight decay 1. We rely on the `HuggingFace Trainer`[1] with default scheduling, `bf16` precision, and `torch_compile` for speed. Training runs up to ∼300k steps **or** a late-phase jump in out-of-distribution (OOD) accuracy emerges. We use three random seeds on a single A100 GPU (48 hours each), reporting the best run (variability ±1–2%).

---

[1] https://github.com/huggingface/transformers

**ID vs. OOD Evaluation.** **In-Distribution (ID)** queries reuse entity/relation combinations observed in training, whereas **Out-of-Distribution (OOD)** queries involve entirely new combinations. The hallmark of "grokking" is a *delayed* surge in OOD accuracy after prolonged training (Power et al., 2022).

### 4.2 Original Dataset (No Augmentation)

Training solely on the original *structured comparison* data produces 100% training accuracy with *no* late-phase OOD jump (Figure 4 (a)).

This plateau aligns with earlier findings that $\phi \approx 0.5$ is insufficient to induce grokking (Wang et al., 2024).

### 4.3 Structured Compositional and Comparison Tasks

**Structured Comparison Tasks.** Figure 4 (b) demonstrates a clear late-phase jump in OOD accuracy when queries ask if two entities share a property (e.g., `City A -- country -- France` vs. `City B -- country -- France`). This simpler relational structure more readily triggers the formation of a generalizing circuit.

**Structured Compositional Tasks.** Figure 4 (c) shows training curves for compositional tasks using triplet-based facts, where chains of the form `X -- spouse of -- Y -- nationality -- Z` are needed.

- **ID Accuracy** rises to near perfection, indicating strong memorization of seen patterns.
- **OOD Accuracy** remains low, showing no late-phase improvement. Complex multi-hop relations appear harder to internalize even with augmentation (Nanda et al., 2023).

### 4.4 Unstructured Tasks

Moving to full Wikipedia paragraphs (*vs.* triplets) adds noise and length (see Figure 4 (d)):

- **Slower Convergence (ID)**: Parsing longer text delays training progress.
- **Modest OOD Gains**: Even with data augmentation, text distractors and ambiguous references limit improvement.

For easier *comparison* queries, the model still attains decent ID accuracy but struggles to generalize OOD, reflecting the sparser effective $\phi$ in unstructured text.

### 4.5 Comparison Table

Table 3 compares our **Grokked GPT-2–small** against GPT-4o and o1-mini on the structured dataset. Our method outperforms others, especially in the comparison task where we reach nearly 100% even in OOD settings. Pretrained models like GPT-4o may not provide a clear ID/OOD distinction since they have seen extensive Wikipedia text.

*Table 3.* **Structured 2WikiMultiHopQA results.** *It is unclear how to provide a clear ID/OOD distinction since models have seen extensive Wikipedia text during training.

| Method | Comparison | | Composition | | Avg | |
|---|---|---|---|---|---|---|
| | ID | OOD | ID | OOD | ID | OOD |
| GPT2-Small | 0.70 | 0.59 | – | 0.03 | 0.70 | 0.31 |
| GPT-4o* | 0.87 | | 0.25 | | 0.56 | |
| o1-mini* | 0.88 | | **0.32** | | **0.60** | |
| **Grokked GPT2-Small** | **1.00** | **0.96** | **0.93** | 0.07 | **0.97** | **0.52** |

### 4.6 Qualitative Analysis

**Success Cases.** When synthetic augmentation covers multi-hop chains (2–3 hops), the model handles queries like *"Which painter was born in the same city as the founder of Company X?"* or *"Do City A and City B share the same country?"* — tasks that otherwise fail at lower $\phi$.

**Failure Cases.** Rare relations or ambiguous entity names still present challenges. If multiple entities share a label but the dataset lacks proper disambiguation, the model may conflate them, hindering OOD performance. Appendix A.4 highlights some examples.

**Overall Findings.** These results confirm that **increasing** $\phi$ via carefully designed synthetic data is key for grokking-based multi-hop QA. While composition tasks and unstructured text may need larger or more targeted augmentation, the "memorization-to-generalization" jump for simpler relational queries demonstrates that *grokking can significantly boost OOD performance* in real-world factual reasoning.

## 5 Limitations and Future Work

There are several avenues for improving and extending our grokking-based approach to multi-hop reasoning. While our proof-of-concept experiments on 2WikiMultiHopQA provide promising evidence, there remain important open questions regarding data complexity, interpretability, and resource feasibility.

**Datasets and Benchmarks.** Our results demonstrate that Transformer grokking can be induced on real-world datasets such as 2WikiMultiHopQA. Nevertheless, a broader spectrum of *challenging* reasoning benchmarks could illuminate the true scope and boundaries of our method. For instance, tasks requiring longer reasoning chains, specialized domain knowledge (e.g., biomedical), or *temporal* reasoning may reveal nuanced constraints that do not emerge in standard Wikipedia-based QA.

**Analysis and Explainability.** Although we observe emergent generalization circuits, the precise *mechanics* of how these circuits form remains only partially understood. Fu-

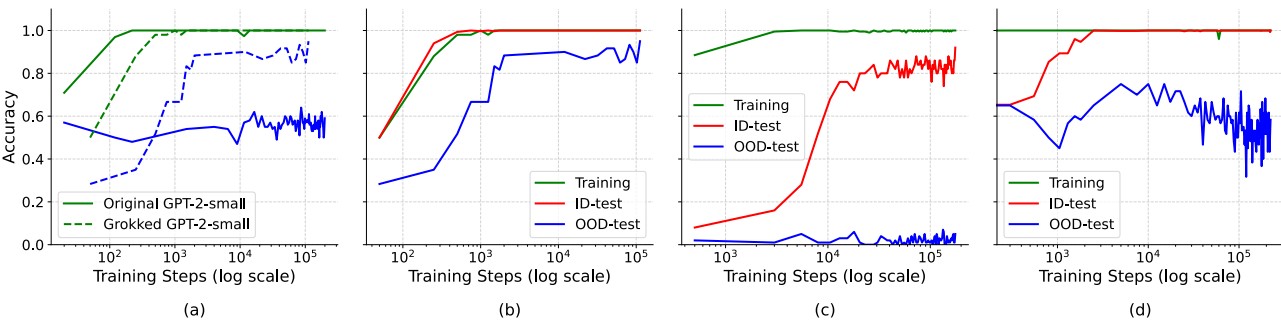

*Figure 4.* **(a) Accuracy on the comparison task for the original and grokked GPT-2-small.** While both models eventually reach perfect training accuracy (green curves), only the grokked GPT-2-small exhibits a significant late-phase improvement in OOD accuracy (blue curves). This indicates that the grokked model continues to learn generalizable structure beyond the training data. **(b) Training curves for the structured comparison task**. ID and OOD behave similarly. **(c) The structured compositional task**. We see near-perfect ID accuracy but no late-phase jump in OOD test accuracy. **(d) Training curves for the unstructured (full paragraph Wikipedia) comparison setting.** Complexity slows convergence and limits OOD gains, although ID accuracy still improves significantly.

ture work can:

- **Quantify factual drift:** Investigate how adding synthetic (hallucinated) facts impacts the model's factual consistency and other downstream metrics.
- **Mechanistic interpretability:** Extend the logit-lens or attention-probing analyses of Wang et al. (2024) to more complex, real-world tasks. Doing so may reveal how sub-networks handle shifting knowledge distributions – particularly if a separate memory module is involved.

**Factuality.** A key question is how synthetic data (some of it intentionally or accidentally hallucinated) affects the model's factual accuracy. While our experiments show that moderate amounts of non-factual data can *bolster* generalization, we acknowledge potential risks:

- **Distortion of real-world knowledge:** Without careful filtering, hallucinations might overwrite or obscure genuine facts.
- **Factual fragility:** Certain tasks (e.g., medical or legal reasoning) demand rigorous correctness, making any factual drift untenable.

As a partial solution, we envision more sophisticated *constraint-based* data augmentation that preserves core factuality while still boosting the inferred-to-atomic ratio $\phi$. Investigating such hybrid strategies is an intriguing direction for future work.

**Scope.** We expanded the scope from contrived toy problems to large-scale, factual datasets derived from Wikipedia. However, there remain many open questions:

- **Non-Wikipedia Domains:** Would the same grokking dynamics hold in domain-specific corpora (e.g., arXiv papers, biomedical literature, or news articles)?
- **Other Reasoning Paradigms:** Beyond factual QA, it is unknown whether a generalizing circuit would also yield improvements on commonsense or moral reasoning tasks,

where the inference rules are less formally grounded.

**Feasibility.** Finally, we note that training large Transformer architectures for extended periods, as required by grokking, can be prohibitively expensive. Techniques to reduce this overhead, such as those described by Lee et al. (2024), are essential. Concretely, future work might explore:

- **Scaling Laws:** Determining how model size, dataset size, and ratio $\phi$ collectively influence training cost.
- **Accelerated Convergence:** Applying curriculum learning or specialized optimizers that shorten the "memorization phase" and expedite the onset of generalization.
- **Pre-training:** Pre-trained models can facilitate grokking by leveraging prior knowledge, improving performance, and accelerating the transition from memorization to generalization. Since they already encode fundamental patterns (e.g., linguistic or mathematical rules), they might require less training time to achieve generalization.

In summary, while we establish the efficacy of grokking for multi-hop factual QA, there is ample room to refine, extend, and **better explain** these emergent capabilities. We hope our work can serve as a foundation for future explorations into more powerful, transparent, and efficient forms of implicit reasoning in large language models.

## 6 Conclusion

We have demonstrated that carefully crafted **data synthesis** can reshape the distribution of factual language corpora in a way that *unlocks* grokking-based generalization. Even a moderately sized GPT-2 model can achieve substantial gains in *multi-hop reasoning* by leveraging the late-phase formation of internal circuits – outperforming more powerful models that do not receive synthetic data augmentation. Moreover, our empirical results indicate that factuality is not significantly compromised; on average, the model's answers become more accurate when given a well-balanced mixture

of real and synthesized facts. The main message of this work is that **boosting the inferred-to-atomic ratio $\phi_r$ via synthetic data remains the most direct route to emergent reasoning circuits**.

Nevertheless, our approach also highlights several **limitations**. Full generalization across all relations requires each relation's atomic facts to be sufficiently augmented, which can be challenging for *rare* or *low-frequency* relations. In many real-world corpora, knowledge graphs are not only sparse but also *disconnected* or partially *non-injective*, limiting the number of multi-hop paths the model can learn from.

Finally, **natural language challenges** persist in practical contexts. Real-world text often contains ambiguous references, unevenly distributed relations, and disjoint subgraphs, making high-quality data augmentation non-trivial. Nonetheless, our work illustrates the promise of *implicit reasoning* – once the ratio of inferred facts surpasses a critical threshold, the model internalizes robust logic circuits that can tackle complex multi-hop queries. We hope these findings encourage further research on efficient synthesis methods, broader domain applications, and deeper analysis of the mechanics behind grokking in large language models.

## Impact Statement

By demonstrating how targeted data augmentation can unlock emergent multi-hop reasoning capabilities, this work paves the way for more robust, interpretable, and efficient knowledge-intensive NLP.

The ability to induce "grokking" on real-world factual datasets promises broader applications, ranging from high-stakes domains (e.g., medical, legal, educational) to everyday question answering.

At the same time, this work underscores the importance of careful curation of synthesized facts to prevent misinformation. This balance between enhanced reasoning and factual accuracy marks a crucial step toward trustworthy, generalizable language models.

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

# A  Appendix

## A.1  Sketch of Proof of Lemma 1

*Proof.* We break the argument into three parts.

**1. Counting all potential $(n+1)$-node sequences.**  A simple *n-hop path* is determined by choosing an ordered tuple of $(n+1)$ distinct entities. The number of ways to choose *distinct* nodes $v_0, v_1, ..., v_n$ out of $|\mathcal{V}|$ is:

$$\binom{|\mathcal{V}|}{n+1}(n+1)!.$$

Here, $\binom{|\mathcal{V}|}{n+1}$ is the number of ways to pick $n+1$ distinct nodes (unordered), and $(n+1)!$ is the number of ways to *order* those nodes into a possible directed path of length $n$.

**2. Probability of each directed path being valid.**  Under the random-graph assumption, the probability that there is a directed edge from $v_{i-1}$ to $v_i$ (for $i = 1, ..., n$) is $\frac{b}{|\mathcal{V}|-1}$. Assuming independence across edges, the probability that *all* $n$ edges are present simultaneously is

$$\left(\frac{b}{|\mathcal{V}|-1}\right)^n$$

**3. Expected number of valid $n$-hop paths.**  By linearity of expectation (applied to each of the $\binom{|\mathcal{V}|}{n+1}(n+1)!$ ordered node tuples), the expected value of $|\mathcal{F}_n|$ is:

$$\mathbb{E}[|\mathcal{F}_n|] = \binom{|\mathcal{V}|}{n+1}(n+1)!\left(\frac{b}{|\mathcal{V}|-1}\right)^n.$$

For large $|\mathcal{V}|$, one can use the binomial approximation $\binom{n}{k} \leq \frac{n^k}{k!}$ for $\binom{|\mathcal{V}|}{n+1}$, such as

$$\binom{|\mathcal{V}|}{n+1} \leq \frac{|\mathcal{V}|^{n+1}}{(n+1)!} \quad \text{(for fixed } n \text{ as } |\mathcal{V}| \to \infty \text{ )}$$

**Computing Bound**  By definition 2.3 and 2.10,

$$\phi_{n,r} = \frac{|\mathcal{F}_{n,r}|}{|\mathcal{F}_{A,r}|} \quad \text{and} \quad |\mathcal{F}_{A,r}| = |\mathcal{V}| \cdot b_r.$$

Together with the result from the third step being $\mathbb{E}[|\mathcal{F}_n|] = \binom{|\mathcal{V}|}{n+1}(n+1)!\left(\frac{b}{|\mathcal{V}|-1}\right)^n$, we obtain,

$$\begin{aligned}
\mathbb{E}[\phi_{n,r}] &= \frac{|\mathcal{F}_{n,r}|}{|\mathcal{V}| \cdot b_r} \\
&= \frac{\binom{|\mathcal{V}|}{n+1}(n+1)!\left(\frac{b_r}{|\mathcal{V}|-1}\right)^n}{|\mathcal{V}| \cdot b_r} \\
&\leq \frac{\frac{|\mathcal{V}|^{n+1}}{(n+1)!}(n+1)!b_r^n}{|\mathcal{V}|b_r(|\mathcal{V}|-1)^n} \\
&= \frac{|\mathcal{V}|^n b_r^{n-1}}{(|\mathcal{V}|-1)^n} \\
&= b_r^{n-1}\left(\frac{|\mathcal{V}|}{(|\mathcal{V}|-1)}\right)^n \\
&= b_r^{n-1}\left(\frac{1}{1-\frac{1}{|\mathcal{V}|}}\right)^n
\end{aligned}$$

Which gives us the asymtotic upper bound,

$$\lim_{|\mathcal{V}|\to\infty} \mathbb{E}[\phi_{n,r}] = \lim_{|\mathcal{V}|\to\infty} b_r^{n-1}\left(\frac{1}{1-\frac{1}{|\mathcal{V}|}}\right)^n \to b_r^{n-1}$$

Hence $\phi_{n,r}$ (and therfore $\phi_n$ overall) remains *boudned above* by $b^{n-1}$, completing the proof.

$\square$

## A.2   Proof of Lemma 2

*Sketch of Proof.* By definition 2.3 and 2.10,

$$\phi_{n,r} = \frac{|\mathcal{F}_{n,r}|}{|\mathcal{F}_{A,r}|} \quad \text{and} \quad |\mathcal{F}_{A,r}| = |\mathcal{V}| \cdot b_r.$$

From A.1, we have

$$|\mathcal{F}_{n,r}| \lesssim \binom{|\mathcal{V}|}{n+1}(n+1)!\left(\frac{b_r}{|\mathcal{V}|-1}\right)^n.$$

because $\frac{b_r}{|\mathcal{V}|-1}$ is the approximate probability that a randomly chosen edge belongs to relation $r$.

Thus,

$$\begin{aligned}
\phi_{n,r} = \frac{|\mathcal{F}_{n,r}|}{|\mathcal{F}_{A,r}|} &= \frac{|\mathcal{F}_{n,r}|}{|\mathcal{V}|\,b_r} \\
&\lesssim \binom{|\mathcal{V}|}{n+1}\frac{(n+1)!\,b_r^{n-1}}{|\mathcal{V}|\,(|\mathcal{V}|-1)^n}.
\end{aligned}$$

**Full generalization** requires $\phi_{n,r} \geq \phi_G$ for *all* $r$ yielding the required constraint

$$\begin{aligned}
&\phi_G \leq \phi_{n,r} \\
\Leftrightarrow\; &\phi_G \leq \binom{|\mathcal{V}|}{n+1}\frac{(n+1)!\,b_r^{n-1}}{|\mathcal{V}|\,(|\mathcal{V}|-1)^n} \\
\Leftrightarrow\; &b_r \geq \sqrt[n-1]{\frac{\phi_G\,|\mathcal{V}|\,(|\mathcal{V}|-1)^n}{\binom{|\mathcal{V}|}{n+1}(n+1)!}}.
\end{aligned}$$

In other words, if *any* relation $r$ has $b_r$ (its branching factor) that fails to exceed this threshold, then $\phi_{n,r}$ cannot reach $\phi_G$. Hence, that particular relation will not "grok," so the $KB$ is *not fully generalizable* over $n$-hop facts (although it might still be *partially* generalizable for other relations). $\square$

## A.3   Formal Derivation of the Node-Count Bound

Similar to A.2, for **full generalization** we can derive a bound for the node count $|\mathcal{V}|$. From the requirement $\forall r \in \mathcal{R}, \phi_{n,r} \geq \phi_G$, we derive,

$$\forall r \in \mathcal{R}, \phi_{n,r} \geq \phi_G$$
$$\Leftrightarrow \min_r \binom{|\mathcal{V}|}{n+1}\frac{(n+1)!b_r^{n-1}}{mathcalV|(|\mathcal{V}|-1)^n} \geq \phi_G$$

in other words, the worst generalizable relation $r$ still needs to fulfill $\phi_{n,r} \geq \phi_G$ for $KB$ to be fully generalizable. Resolving after $|\mathcal{V}|$ without the use of the binomial approximation yields,

$$\min_r \binom{|\mathcal{V}|}{n+1} \frac{(n+1)!b_r^{n-1}}{|\mathcal{V}|(|\mathcal{V}|-1)^n} \geq \phi_G$$

$$\Leftrightarrow \min_r \frac{|\mathcal{V}|!}{(n+1)!(|\mathcal{V}|-n-1)!} \frac{(n+1)!b_r^{n-1}}{|\mathcal{V}|(|\mathcal{V}|-1)^n} \geq \phi_G$$

$$\Leftrightarrow \min_r \frac{|\mathcal{V}|!}{(|\mathcal{V}|-n-1)!} \frac{b_r^{n-1}}{|\mathcal{V}|(|\mathcal{V}|-1)^n} \geq \phi_G$$

$$\Leftrightarrow \frac{(|\mathcal{V}|-1)!}{(|\mathcal{V}|-n-1)!(|\mathcal{V}|-1)^n} \min_r b_r^{n-1} \geq \phi_G$$

$$\Leftrightarrow \frac{(|\mathcal{V}|-1)!}{(|\mathcal{V}|-n-1)!(|\mathcal{V}|-1)^n} \geq \max_r \frac{\phi_G}{b_r^{n-1}}$$

$$\Leftrightarrow \frac{\Gamma(|\mathcal{V}|)}{\Gamma(|\mathcal{V}|-n)(|\mathcal{V}|-1)^n} \geq \max_r \frac{\phi_G}{b_r^{n-1}}$$

Thus we have:

$$|\mathcal{V}| \geq \min\left\{v \in \mathbb{N} : \frac{\Gamma(v)}{\Gamma(v-n)(v-1)^n} \geq \max_r \frac{\phi_G}{b_r^{n-1}}\right\}.$$

For an empirical example (i.e., using a randomly generated graph), see Figure 5.

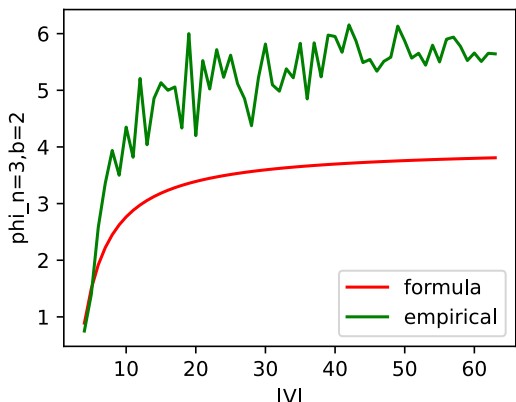

*Figure 5.* Growth of $\phi_{3,r}$ (**y-axis**) with $|\mathcal{V}|$ (**x-axis**) for $b_r = 2$. The **red line** is $\phi_{3,r} = \binom{|\mathcal{V}|}{4} \frac{96}{|\mathcal{V}|(|\mathcal{V}|-1)^3}$ (formula values). The **green line** are empirical values obtained by randomly generating a graph with the same amount of nodes ($|\mathcal{V}|$) and edges ($|\mathcal{V}|b_r$). Due to randomness, our generated/empirical graph can have locally higher branching factors, resulting in overall more inferred facts. Our formula for $\phi_{3,r}$ therefore effectively underestimates the true value of $\phi_{3,r}$. Nevertheless, the formula correctly approximates the shape and order of magnitude. This holds also for other combinations of $b_r$ and $n$.

## A.4 Qualitative Examples

In the following section, we present qualitative examples of QA pairs (*"Question", "Ground Truth"*) from the composition and comparison tasks that the model failed to classify correctly. Some errors stem from inconsistencies in the 2WikiMultiHopQA dataset, while others arise due to our augmentation strategy. Overall, we believe that the model is capable of achieving 100% accuracy, as demonstrated in previous studies.

**Composition** There are several grammatical inconsistencies in the 2WikiMultiHopQA dataset that prevent our model from reaching 100% accuracy.

**Nationality questions may have inconsistent ground truth formats** For nationality-related questions, the ground truth can be either an adjective or the name of a country, even when the latter is grammatically incorrect.

**Question:** What nationality is William Seymour, 3rd Duke of Somerset's father?
**Ground Truth:** English

**Question:** What nationality is Amadeus VIII, Duke of Savoy's mother?
**Ground Truth: France**

**Adjective instead of noun in country-related answers**   In some cases, the ground truth is an adjective instead of a noun referring to a country.

**Question:** Which country is the trumpeter of Paper Bird from?
**Ground Truth:** American

**Comparison** Comparison errors are primarily due to limitations in the data augmentation algorithm, which failed to generate a sufficient number of inferred facts for some relational queries. Most of them are related to lakes, rivers, or airports, which we attribute to the sparsity of the base data.

**Question:** Are both Wainwright/Wainwright Field 21 Airport and Roberval Air Saguenay Water Aerodrome located in the same country?
**Ground Truth:** Yes

**Question:** Are Long Lake, East Ferris, Ontario, and Montreal Lake, Saskatchewan, both located in the same country?
**Ground Truth:** Yes

**Question:** Are Deer Creek, Osage River, and Big Prairie Dog Creek both located in the same country?
**Ground Truth:** Yes

**Question:** Are Chicoutimi/Saint-Honoré Aerodrome and Rtishchevo Air Base both located in the same country?
**Ground Truth:** No

## A.5   Data Synthesis

Here, we present the prompts used throughout our data augmentation process. During our experiments, we leveraged both GPT-4o and o1-mini to generate diverse and high-quality augmented data.

### A.5.1   COMPOSITION

**Graph parsing**

```
You are graph gpt.  You build graph based on the provided text.  Find all objects, their
relations and types.
Pick one of the following types:  - Person - Location - Object (include everything that
was not above)
Return the following format with numbering:  1.  <Avatar; Film><director><James Cameron;
Person> 2.  <James Cameron; Person><directed><Titanic; Object>
```

**Question formatting**

```
You are a question formatting assistant.  Your task is to create questions based on the
given relations and objects.
Use the provided examples as a guide for the question style.  Ensure that the answer
remains unchanged and enclosed in <a> tags.  You may rephrase one question, given the
example format.  Strictly follow the logic of given examples.  Connect it in the following
logic:  <obj1> -> <rel1> -> <rel2> -> <obj3>
Return numbered responses in format:  1.  What is the director of the film that James
Cameron produced?<a>Steven Spielberg</a> 2.  Who directed the movie starring Tom
Cruise?<a>Christopher Nolan</a>
```

A.5.2  COMPARISON

**Atomic fact generation**

```
You are a helpful assistant that generates geographical facts.  Generate new unique
locations and their countries in the following format:  Follow the style of the examples,
but do not use the same locations.
Rules:  1.  Use real locations and countries 2.  Each location should be unique 3.  DO
NOT REUSE PROVIDED EXAMPLES 4.  Do not answer the question – only provide locations 5.  Do
not use formatting except for numbering 6.  Generate equal amount of NEW!!!  locations for
following countries:
```

**Detailed atomic fact generation**

```
You are a helpful assistant that generates geographical facts.  Based on the provided
examples, generate a paragraph for each location-country pair.  Strictly follow the
style and lenght of the provided examples Do not answer the question – only provide the
paragraph with numbering.  DO not return empty lines.  One by one.  Return the number
according to the given data.  Here are the examples:
```

