# OpenReview forum: "Grokking in the Wild: Data Augmentation for Real-World Multi-Hop Reasoning with Transformers"
_ICML.cc/2025/Conference — ICML 2025 poster_

### Official Review · Reviewer_TzCM · 2025-03-12

**Overall Recommendation:** 3

**Summary:**

This paper examines the problem of learning multi-hop reasoning over knowledge graph facts. Prior works have shown that this is a challenging problem, particularly without chain-of-thought or externalized reasoning. However, recently, synthetic experiments have revealed that such multi-hop reasoning can be learned by models in a grokking regime -- provided that there exists a sufficient concentration of inferred facts in the dataset. This paper examines whether such a grokking phenomenon occurs in a real fact-learning setting. They first show theoretically that there exists a lower bound of the knowledge graph branching factor which determines whether  there are "naturally" a sufficient number of inferred facts to enable reliably learning multi-hop reasoning. In real world settings, based on the WikiData, the authors demonstrate that there do not exist a sufficient concentration of inferred facts and they describe possibilities of generating additional data to boost the formation of multi-hop reasoning. Empirically, they show that training on this augmented data does in fact induce grokking dynamics in real transformers. Impressively, they can compare with very small models and achieve comparable performance to the largest models.

**Claims And Evidence:**

The main claim of this paper is that grokking can emerge in real settings (esp. for multi-hop fact reasoning settings). This is tested via a set of experiments on Wikipedia-based knowledge graph settings. They have two settings: a structured knowledge setting in which facts are represented as triples. In particular, this is where they observe the Grokking phenomena primarily. However, I believe that the structured setting eliminates much of the complexity and noise present in real fine-tuning datasets. On the other hand, they also consider an unstructured setting. In this setup, however, the results do not appear to confirm grokking as far as I can tell. Thus, I am not sure that this claim is necessarily supported.

**Essential References Not Discussed:**

There is not one that I am aware of.

**Experimental Designs Or Analyses:**

I did not have specific concerns about the validity of the experimental design, with the exception of my question about in-distribution/out-of-distribution accuracy above. Also for clarity, how exactly was Grokked GPT-2–small trained. Was it trained on the unstructured data or the structured data? Additional details of the prompts tried for the comparison to GPT-4/o1 would be quite helpful to better contextualize the performance of the proprietary models..

**Methods And Evaluation Criteria:**

The 2-Wiki-Multi-Hop dataset appears to be a reasonable setting to test and the model size is reasonable -- albeit maybe on a smaller end for real tasks. It would be interesting if the authors had some predictions about the impact of model size on their results. I also found that the discussion of in-distribution/out-of-distribution were a bit confusing. The authors state that in-distribution performance measures entity pairs seen during training. If that is the case, how does this differ from the training loss?

**Other Comments Or Suggestions:**

Line 192: "you sure you wanted to change that to a ”not exists” because" -- Is this a typo?

**Other Strengths And Weaknesses:**

I believe that this paper could benefit significantly from better presentation. Overall, the problem studied, experiments, and theoretical arguments are interesting in my opinion but they are hard to parse. In the theory section, the authors introduce a large amount of formalism and notation. I find that this overwhelms me with many details (for example they introduce a norm over edges but this norm isn't used later in the text as far as I can tell). I  think the authors could make this paper significantly easier to follow by culling the notation and excessive detail in the theory section and deferring some notions to the supplementary material.

**Questions For Authors:**

1) Please answer the questions in "Experimental Design or Analysis"

**Relation To Broader Scientific Literature:**

This paper appears to borrow heavily from the formulation/problem setup etc of [1]. However, they extend the simulated/toy settings addressed by [1] and instead consider real knowledge graph data -- sourced from WikiData. Their claimed key contribution is that they show the phenomena introduced by [1] can in fact happen in practice. I will note that in the structured setting (the place where they see the  most direct example of grokking) the problem is actually not so different from the toy settings examined by [1].





[1] Wang, B., Yue, X., Su, Y., and Sun, H. Grokked Transformers are Implicit Reasoners: A Mechanistic Journey to
the Edge of Generalization, May 2024. URL http://
arxiv.org/abs/2405.15071. arXiv:2405.15071
[cs].

**Theoretical Claims:**

I have briefly reviewed the theoretical sections, but not the proofs. The theory appears to be predicated on the necessity of observing a given number of multi-hop paths. The main theoretical result proves a lower bound on the average branching factor of the knowledge graph necessary to achieve generalization. I think this analysis is generally ok, but it does assume the existence of some "known" generalization ratio for each relation. It would be better if the authors can justify why it is ok to model this generalization ratio as only depending on the relation. Could it not also depend on the coverage of the entities in the seen multi-hop examples?

---

> ### Author Rebuttal · Authors · 2025-03-31
>
> We thank the reviewer for the thoughtful and detailed feedback, which has helped us refine both the presentation and scope of the paper. Below, we address each of the raised concerns.
>
> **A. On theoretical presentation and unused formalisms:**
>         We appreciate your observation regarding the mathematical complexity and notation density in the theory section. In response, we have carefully reviewed all definitions and formal constructs. A dependency analysis shows that each definition contributes to Lemma 1 or 2, either directly or through intermediate derivations. While we agree that the presentation could be simplified, we found that deferring key definitions to the appendix would compromise the readability and cohesion of the main theoretical argument. We will, however, streamline redundant notation (e.g., the unused edge norm) and improve clarity in the final version. Thank you also for catching the typo on line 192; we have corrected it.
>
> **B. On in-distribution vs. training data distinction:**
>         Thank you for highlighting the confusion here. We will clarify this distinction in the final version. In short: the training data consists of atomic facts and associated inferred facts used in specific combinations. In-distribution (ID) evaluation involves novel combinations of *seen atomic facts*, not seen together during training. For example, if the model saw the atomic facts for "Avignon Rocher des Doms" and "Paris Louvre Museum" in training, an ID question might involve "Avignon Rocher des Doms" and the "Eiffel Tower", a new combination of familiar elements. This contrasts with out-of-distribution (OOD) examples, which include entirely new atomic facts.
>
> **C. On structured vs. unstructured settings and realism:**
>  We agree that structured settings simplify linguistic complexity. Our core claim is not that we have solved the problem for all real-world unstructured data, but that the grokking phenomenon (previously limited to toy datasets) can occur with real-world entities and linguistic structure, provided key conditions are met (notably a sufficiently high $\phi_r$). The structured format allowed us to isolate this effect. In the unstructured setting, generalization circuits are harder to induce due to syntactic variability and ambiguity. We view extending grokking to unstructured data as an important avenue for future work. Our current contribution lies in demonstrating that grokking is not limited to synthetic data, and that it can emerge from real-world facts when structured and augmented appropriately.
>
> **D. On relation-specific generalization ratios:**
>         The assumption that $\phi_r$ can be modeled as relation-specific follows from prior work [1], which demonstrates that transformer-based generalization circuits rely primarily on relation-specific atomic facts. This behavior has been empirically confirmed in our own replications of their experiments. While entity frequency affects the number of available paths, it is the *relation* that governs the type and reusability of inference patterns, making $\phi_r$ effectively relation-dependent. We clarify this distinction in Lemma 1 and Appendix A.1, where we show that while the number of unique paths grows with the number of entities $|\mathcal{V}|$, the asymptotic behavior is dominated by the branching factor $b^{n-1}$, not by entity coverage.
>
> **E. On model size effects:**
>         We are currently conducting experiments with varying GPT-2 sizes (124M–1.5B) to investigate the impact of scale on generalization dynamics. Early results suggest that larger models reach generalization faster, but the critical $\phi_r$ threshold remains stable. For additional details, please see our response to Reviewer hZJa (Section A).
>
> **F. On prompt transparency and training details:**
>         Thank you for raising this point. We will include full prompt templates, sampling parameters, and training details for both structured and unstructured settings in the final version. Briefly, we trained two separate GPT-2 small models from scratch—one for structured data (triple format) and one for unstructured data (natural language questions). Neither model used pretraining; both were trained solely on task-specific data.
>
> **G. On circuit flexibility and multi-hop reasoning:**
>         The generalization circuits observed in our setting are relatively rigid and rely on deterministic retrieval and composition of atomic facts. For 2-hop reasoning, this involves a single inference operation; for $n$-hop paths, multiple sequential inference steps are required. Based on our theoretical framework and ongoing experiments, we are confident that $n$-hop grokking is feasible without fundamental changes to the architecture -- provided the augmented dataset supports sufficient multi-hop path coverage.
>
> We again thank the reviewer for the constructive feedback and will incorporate these improvements in the final submission.

---

### Official Review · Reviewer_vnAC · 2025-03-14

**Overall Recommendation:** 3

**Summary:**

The paper investigates the application of grokking—a phenomenon where neural networks transition from memorization to generalization after prolonged training—to real-world multi-hop reasoning tasks. The authors propose augmenting sparse knowledge graphs (KGs) with synthetic data to increase the ratio of inferred-to-atomic facts ($\phi_r$), enabling Transformers to form internal reasoning circuits. Experiments on the 2WikiMultiHopQA benchmark show that their approach achieves near-perfect accuracy (95–100%) on comparison tasks, outperforming baselines and matching state-of-the-art models. The paper also provides mechanistic insights into how increasing $\phi_r$ drives circuit formation in Transformers.

**Claims And Evidence:**

1. Synthetic data augmentation raises $\phi_r$ above a threshold, enabling grokking in real-world KGs.  Supported by experiments showing that increasing $\phi_r$ via synthetic multi-hop paths correlates with late-phase generalization (Figure 3).
2. Even factually incorrect synthetic data can improve reasoning by forcing reliance on relational structure.  Partially supported by qualitative analysis showing improved OOD accuracy despite synthetic noise. However, no explicit ablation studies compare factual vs. non-factual synthetic data.
3. Transformers trained with this method achieve SOTA results on 2WikiMultiHopQA.  Validated by results in Table 3, where their GPT-2-small model outperforms GPT-4o and o1-mini on structured tasks.

**Essential References Not Discussed:**

N/A

**Experimental Designs Or Analyses:**

**Strengths**:
- Clear ID/OOD splits to measure generalization.
- Multiple seeds and architecture details ensure reproducibility.

**Weaknesses**:
- **Synthetic Data Generation**: The process relies on LLM prompting but lacks transparency (e.g., prompts not fully disclosed).
- **Baselines**: Comparisons to GPT-4o are unclear due to potential data leakage (GPT-4 may have seen 2WikiMultiHopQA during pretraining).

**Methods And Evaluation Criteria:**

**Methods**:
- **Data Synthesis**: Augment KGs with LLM-generated atomic/inferred facts to boost \(\phi_r\).
- **Training**: Train GPT-2-style Transformers with prolonged optimization (300k steps) to induce grokking.

**Evaluation**:
- **Benchmark**: 2WikiMultiHopQA (structured/unstructured subsets).
- **Metrics**: In-distribution (ID) vs. out-of-distribution (OOD) accuracy.

**Strengths**:
- The focus on $\phi_r$ as a key lever for grokking is novel and grounded in prior theoretical work.
- Evaluation across both structured (triplet-based) and unstructured (paragraph-based) settings adds robustness.

**Weaknesses**:
- No comparison to retrieval-augmented methods, which are common in multi-hop QA.

**Other Comments Or Suggestions:**

N/A

**Other Strengths And Weaknesses:**

**Strengths**:
- First to apply grokking to real-world KGs.
- Mechanistic analysis of circuit formation.

**Weaknesses**:
- Limited scalability analysis (e.g., larger KGs or models).
- Risks of synthetic hallucinations are acknowledged but not quantified.

**Questions For Authors:**

How does the required $\phi_r$ threshold vary with model size?

**Relation To Broader Scientific Literature:**

The work bridges grokking[1] and multi-hop QA [2,3]. It extends grokking beyond synthetic tasks, aligning with efforts to study emergent reasoning in Transformers[4]. However, it does not discuss connections to neuro-symbolic methods (e.g., logical reasoning modules).


- [1] Grokking: Generalization beyond overfitting on small algorithmic datasets
- [2] HotpotQA: A Dataset for Diverse, Explainable Multi-hop Question Answering
- [3] Constructing A Multi-hop QA Dataset for Comprehensive Evaluation of Reasoning Steps.
- [4] Grokked Transformers are Implicit Reasoners: A Mechanistic Journey to the Edge of Generalization,

**Theoretical Claims:**

The paper builds on grokking theory[1] but does not introduce new theoretical proofs. Instead, it adapts existing concepts (e.g., generalization thresholds for $\phi_r$) to real-world KGs. The lemmas on asymptotic bounds for multi-hop paths (Appendix A.2–A.4) are intuitive but lack formal proofs.

[1] Grokking: Generalization beyond overfitting on small algorithmic datasets

---

> ### Author Rebuttal · Authors · 2025-03-31
>
> We thank the reviewer for the detailed and constructive feedback.
>
> **A. On transparency and reproducibility:**
>         We appreciate the emphasis on methodological clarity. In the final version, we will provide full details of the training procedure, including the exact prompts used for structured and unstructured data synthesis, temperature and sampling parameters, and other relevant hyperparameters. We will also include the target responses used for comparison and make code and data available to support reproducibility.
>
> **B. On comparisons with large pre-trained models:**
>         We agree that comparing against models such as GPT-4o presents challenges due to potential training data leakage, especially with benchmarks like 2WikiMultiHopQA, which overlap with common pretraining corpora such as Wikipedia. We note that this limitation affects most publicly available models and motivates our decision not to report IID/OOD splits for those baselines. Nonetheless, the fact that our grokked GPT-2-small model surpasses these models in structured comparison tasks underscores the potential of $\phi_r$-driven grokking as a viable, lightweight alternative for reasoning over factual knowledge.
>
> **C. On model scaling and $\phi_r$ thresholds:**
>         We are currently conducting experiments across a range of GPT2 model sizes (124M–1.5B parameters) to evaluate the effect of scaling on generalization dynamics. Preliminary results suggest that while grokking behavior remains consistent, larger models tend to reach generalization more rapidly. Notably, the critical $\phi_r$ threshold required for generalization appears stable across model sizes. Please also refer to our response to Reviewer hZJa (Section A) for more details.
>
> **D. On theoretical rigor:**
>         We are in the process of finalizing complete versions of the theoretical results, including clearly stated assumptions and formal proofs to replace the current sketches in Appendix A.2–A.4. These revisions will be included in the final version.

---

### Official Review · Reviewer_hZJa · 2025-03-17

**Overall Recommendation:** 4

**Summary:**

This paper explores extending the phenomenon of "grokking"—where neural networks transition from memorization to generalization after prolonged training—from synthetic tasks to real-world factual reasoning. The authors address the challenge of dataset sparsity in real-world knowledge graphs by proposing a data augmentation strategy to increase the ratio ($\phi$) of inferred facts to atomic facts beyond the threshold required for grokking to emerge. This augmentation process includes the addition of both factually correct and, in some cases, incorrect synthetic data, with the latter aiming to encourage reliance on relational structures rather than memorization.
The paper evaluates its approach on the 2WikiMultiHopQA benchmark, focusing on multi-hop reasoning tasks. Results show that a GPT2-small model achieves up to 95-100% accuracy on comparison tasks after grokking occurs.
The paper also provides a formal theoretical framework to define necessary conditions for knowledge graphs to be "generalizable," focusing on the branching factor of relations. These findings suggest that enhancing the inferred-to-atomic fact ratio is a critical factor for enabling robust generalization circuits in transformers. The authors acknowledge that while the benchmarks demonstrate improvements, further investigation is needed to evaluate the applicability of these methods to other domains and real-world scientific discovery tasks.

**Claims And Evidence:**

The primary claims -- that grokking can be extended from synthetic to real-world factual reasoning, that the $\phi$ ratio threshold provides a figure of merit for grokking, and the theoretical bounds necessary for generalization -- are well supported by the work.  The late-phase out-of-distribution jump demonstrated in Figure 3-b illustrates the general premise, and considerable attention is given to the formalism and bounds.

**Essential References Not Discussed:**

None.

**Experimental Designs Or Analyses:**

The experimental design effectively isolates the impact of the $\phi$ ratio on grokking and provides valuable comparisons across structured/unstructured formats. Minor weaknesses include testing only on GPT2-small and the lack statistical significance.

**Methods And Evaluation Criteria:**

The paper uses data augmentation to increase the ratio of inferred-to-atomic facts in knowledge graphs, testing on 2WikiMultiHopQA across structured/unstructured formats. In-distribution and out-of-distribution evaluations are used to quantitatively assess generalization capabilities, and experimental evidence aligns with theoretical results.

**Other Comments Or Suggestions:**

None.

**Other Strengths And Weaknesses:**

None.

**Questions For Authors:**

None that are fair to ask, as I think they are follow-on work to this paper -- there are a lot of interesting questions about the $\phi$ scaling with architecture and model size that I hope to see the authors pull the thread on in the future.

**Relation To Broader Scientific Literature:**

I see no missing material on the relation of the papers results to broader scientific literature.

**Theoretical Claims:**

The paper presents two key theoretical claims: 1) an asymptotic bound on the number of n-hop paths in knowledge graphs as a function of entities, branching factor, and edge probability; and 2) necessary conditions for a knowledge graph to be "generalizable" based on relation-specific branching factors.  The sketched proofs seem like they could be made more rigorous with more clearly articulated assumptions, but they provide sufficiently compelling intuition for the paper -- I'm not sure whether "Lemma" is the appropriate word for the results.

---

> ### Author Rebuttal · Authors · 2025-03-31
>
> We thank the reviewer for the thoughtful and constructive feedback.
>
> **A. On model size and generalization:**
>  We appreciate the suggestion regarding scaling effects. We are currently conducting additional experiments across GPT2 model sizes (124M to 1.5B parameters). Preliminary results indicate that the grokking behavior is qualitatively consistent across sizes, especially on structured data. However, larger models tend to reach generalization more quickly. We will include these findings, along with supporting graphs and analysis, in the final version. Notably, the critical ratio $\phi$ remains stable across model sizes.
>
> **B. On applicability to other domains:**
>         We agree that extending this work to broader domains is an important next step. We are actively exploring applications beyond fact-based reasoning, including logic-intensive tasks such as mathematical problem solving, and real-world scientific domains. These follow-up studies aim to evaluate the generality of $\phi$-driven generalization. We believe this line of inquiry could inform the design of data-efficient training strategies in broader AI applications.
>
> **C. On theoretical rigor:**
>         Thank you for the feedback on the formal results. We are currently refining the theoretical section to provide complete, rigorous proofs with clearly stated assumptions. The final version will replace the current sketches with detailed derivations, and we will revise the terminology (e.g., replacing “Lemma” if more appropriate) to ensure precision. We will also discuss the limitations and scope of the theoretical claims to better contextualize the results.

---

### Decision · Program_Chairs · 2025-05-01

**Decision:**

Accept (poster)

**Comment:**

The paper presents some interesting insights on how generalization may happen in language models' training. It argues that inferred-facts ratio should be higher and present a data augmentation method to achieve such a training set. Reviewers appreciated the insights presented in the paper, both from a theoretical and practical aspect. However, reviewers also noted that theory section could have had more details (beyond proof sketches) and the writing could have been clearer.
On balance, I think the paper makes a good contribution and applies to the real-world usecase of reasoning over facts.